# From Modified Newtonian Dynamics to Superfluid Vacuum Theory

**DOI:** 10.3390/e25010012

**Published:** 2022-12-21

**Authors:** Tony C. Scott

**Affiliations:** Institut für Physikalische Chemie, RWTH Aachen University, 52056 Aachen, Germany; tcscott@gmail.com

**Keywords:** MOND, dark matter, superfluids, dilatons, relativity, Everett–Hirschman entropy, logarithmic Schrödinger equation

## Abstract

Herein is a review of the essentials of Modified Newtonian Dynamics (MOND) versus dark matter models based on Superfluids for modeling galactic rotation curves. We review the successes and issues of both approaches. We then mention a recent alternative based on the Superfluid Vacuum Theory (SVT) with a nonlinear logarithmic Schrödinger equation (LogSE) which reconciles both approaches, retains the essential success of MOND and the Superfluid nature but does not necessitate the hypothesis of processes including dark matter. We conclude with the implications of this SVT alternative on quantum theory itself.

## 1. Introduction

Because we have been burdened with the notions of dark matter (DM) and dark energy (DE), and because matter is claimed to be only 5% of the Universe, making us the very outliers of existence, it is not surprising that we have to be imaginative when it comes to modeling the Universe. The leading paradigm of DM already suggests it is extremely cold (as in the cold of outer space), which indicates that DM, or rather cold dark matter (CDM), could be a viable Superfluid, meaning a fluid with no viscosity with a temperature near absolute zero.

Nonetheless, Modified Newtonian Dynamics (MOND), a theory that proposes a modification of Newton’s laws to account for the observed properties of galaxies, has been very successful for cosmological studies. It is an alternative to the theory of DM in terms of explaining why galaxies do not appear to obey the currently understood laws of physics.

As aptly pointed out by Sabine Hossenfelder [1], MOND can:Obtain the correlation between mass and rotational velocity giving rise to the observed flat curves on the outskirts of galaxies;Avoid galaxy cusps;Reduce the number of dwarf galaxies;Help model the planar arrangement of satellite galaxies.

On galactic scales, MOND is simpler than DM models and more predictive. However, MOND cannot model the early universe nor galaxy clusters. MOND started as a non-relativistic theory and even its relativistic generalizations do not emerge simply from general relativity. MOND’s success is undeniable, but it is only a useful approximation. Its greatest weakness is that it cannot account for what DM models can. Thus, we review MOND, its relativistic generalization, and DM models and consider the possibility of getting the best of all possible worlds to model galactic rotation curves.

This work is outlined as follows. We start with a review of MOND, build up to the Superfluid Lagrangian formulations and finally lead up to the relativistic DM Lagrangian formulation of Khoury et al. We mention the pros and cons of these approaches and then present our own alternative based on the log Bose–Einstein condensate (BEC) Superfluid Vacuum Theory (SVT) involving the logarithmic Schrödinger equation (LogSE). We mention its properties and desirable features for modeling not only BECs such as Superfluids but also galactic rotation curves. Finally, we conclude with some comments regarding its implications on quantum theory. Note that apart from the abbreviations, each distinct section follows its own notation, largely that of the authors the section represents.

## 2. MOND Preliminaries

According to Sabine Hossenfelder [1], the main potentials for Modified Newtonian Dynamics (MOND) are given by Equation (1):(1)$$\begin{array}{cc} \mathrm{Newtonian}\mathrm{Gravity} & \mathrm{Modified}\mathrm{Newtonian}\mathrm{Gravity}\\ \; & \\ \Phi =-\frac{MG}{r} & \Phi =\sqrt{MGa_{0}}\ln\left(\frac{r}{MG}\right)\\ F=\frac{MG}{r^{2}} & F=\frac{\sqrt{MGa_{0}}}{r} \end{array}$$

### 2.1. Significance of the MOND Acceleration Constant

As best stated by Sabine Hossenfelder: “The acceleration scale that best fits the data turns out to be related to the cosmological constant. No one has any idea why” [2].
(2)$$a_{0}\approx \sqrt{\frac{\Lambda }{3}}$$One has to be careful here as mathematicians often set fundamental constants like the speed of light to unity in general relativity (GRT) within cosmological studies and therefore the scaling is lost. It should be recognized that the RHS of Equation (2) is proportional to the Hubble constant in the t→∞ limit in Eddington’s cosmology ([3], p.193). It is also seen in the *Friedmann–Robertson–Walker* (FRW) model, one of the first successful GRT models, an exact solution for a homogeneous, isotropic and expanding universe. These latter properties stand out as perhaps outrageous approximations and assumptions [4].

This realization, as well as an explanation for why this is a good fit, is given by one of the pioneers of MOND, i.e., M. Milgrom himself [5], as given by these two successive quotes:

“$a_{0}$ can be determined from several of the MOND laws in which it appears, as well as from more detailed analyses, such as of full rotation curves of galaxies. All of these give consistently $a_{0}\approx \left(1.2\pm 0.2\right)\times 10^{-8}$ cm $s^{-2}$. It was noticed early on [6,7,8] that this value is of the order of cosmologically relevant accelerations:
$${\bar{a}}_{0}=2\pi a_{0}=cH_{0}=c^{2}\sqrt{\frac{\Lambda }{3}}\;,$$ where $H_{0}$ is the Hubble constant, and Λ the cosmological constant. In other words, the MOND length, $\ell_{M}\approx 7.5\times 10^{28}$ cm $\approx 2.5\times 10^{4}$ Mpc, is of order of today’s Hubble distance, namely, $\ell_{M}\approx 2\pi \ell_{H}\;\left(\ell_{H}\equiv c/H_{0}\right)$, or of the de Sitter radius associated with Λ, namely, $\ell_{M}\approx 2\pi \ell_{S}$. The MOND mass, $M_{M}\approx 10^{57}$ gr, is then $M_{M}\approx 2\pi c^{3}/GH_{0}\approx 2\pi c^{2}/G{\left(\Lambda /3\right)}^{1/2}$, of the order of the closure mass within today’s horizon, or the total energy within the Universe observable today.”

“For example, by the heuristic idea put forth in [9], it is the quantum vacuum–which is shaped by the state of the Universe–that is the inertia-giving agent. The origin of $a_{0}$ in cosmology also emerges, and is indeed $∼c^{2}\Lambda^{1/2}$. The vacuum then serves as an absolute inertial frame (acceleration with respect to the vacuum is detectable, e.g., through the Unruh effect). Here, it is cosmology that enters local dynamics to give rise to the MOND-cosmology coincidence. The “interpolating function” is not put in by hand, but emerges. It could be calculated only for the very special (and impractical) case of eternally constant, linear acceleration, *a*. If we generalize Newton’s second law to F=mA(a), then one finds $A\left(a\right)={\left(a^{2}+c^{4}\Lambda /3\right)}^{1/2}-{\left(c^{4}\Lambda /3\right)}^{1/2}$. At high accelerations, $a\gg {\left(c^{4}\Lambda /3\right)}^{1/2}$, it gives the Newtonian expression, A=a, while at low accelerations, $a\ll {\left(c^{4}\Lambda /3\right)}^{1/2}$, we have $A=a^{2}/{\left(4c^{4}\Lambda /3\right)}^{1/2}$. This is exactly the required MOND behavior; furthermore, the observed relation $a_{0}∼{\left(c^{4}\Lambda /3\right)}^{1/2}$ is gotten. The “interpolating function” underlying this result is analogous to the “interpolating function” that enters the relation between the kinetic energy, $E_{K}$, and the momentum P and M should be in italic, *P*, of a particle of mass *M*, in special relativity: $E_{K}=E-Mc^{2}={\left(P^{2}c^{2}+M^{2}c^{4}\right)}^{1/2}-Mc^{2}$, which in the limit P≫Mc gives $E_{k}=Pc$, and at low momenta, P≪Mc, gives $E_{K}=P^{2}/2M$.”

The “interpolating function” is explained in the next section which shows the simple fundamentals of the MOND theory.

### 2.2. Simple Non-Relativistic MOND Theory

The Poisson action for the gravitational potential, which dictates particle accelerations according to a=−∇ϕ, is replaced by this nonlinear version [10]:(3)$$\nabla \cdot \left[\mu \left(\frac{∥\nabla \phi ∥}{a_{0}}\right)\nabla \phi \right]=4\pi G\rho \;,$$
where ϕ is the standard Newtonian gravitational potential and
(4)$$\begin{array}{ccc} \mu (x)\to 1 & & for\;x\gg 1\;,\\ \mu (x)\to x & & for\;x\ll 1\;, \end{array}$$
where μ(x) is the “interpolating function”. Two common choices for μ are:(5)$$\begin{array}{ccc} \mu \left(\frac{a}{a_{0}}\right) & = & \frac{1}{1+\frac{a_{0}}{a}}\;,\\ \mu \left(\frac{a}{a_{0}}\right) & = & \sqrt{\frac{1}{1+{\left(\frac{a_{0}}{a}\right)}^{2}}}\;, \end{array}$$
where *a* is an acceleration, and $a_{0}$ is a new fundamental constant which marks the transition between the Newtonian and deep-MOND regimes. Equation (3) can be solved given suitable boundary conditions and the choice of μ to yield Milgrom’s law (up to a curl field correction which vanishes in situations of high symmetry). This law, the keystone of MOND, is chosen to reduce to the Newtonian result at high acceleration but leads to different (“deep-MOND”) behavior at low acceleration. In the deep-MOND regime, $\left(a\ll a_{0}\right)$:(6)$$F_{N}=m\mu \left(\frac{a}{a_{0}}\right)a=m\frac{a^{2}}{a_{0}}.$$Here, $F_{N}$ is the Newtonian force and *m* is the object’s (gravitational) mass. When the object is in circular orbit around a point mass *M* (a crude approximation for a star in the outer regions of a galaxy), we find:(7)$$\frac{GMm}{r^{2}}=m\frac{{\left(\frac{v^{2}}{r}\right)}^{2}}{a_{0}}\;⟹\;v^{4}=GMa_{0}\;,$$
i.e., the star’s rotation velocity is independent of *r*, its distance from the center of the galaxy, and the rotation curve is flat, as required. This is the justification behind the MOND formalism: a phenomenological modification of Newton’s law specifically designed to produce flat rotation curves. By fitting his law to rotation curve data, Milgrom found $a_{0}\approx 1.2\times 10^{-10}\;{\mathrm{ms}}^{-2}$ to be optimal. This simple law is sufficient to make predictions for a broad range of galactic phenomena.

In the case of spherical symmetry, the solution to Equation (3) in the MOND regime (certainly for the first case of μ(x) is $\Phi_{MOND}∼\ln\left(r\right)$, as opposed to $\Phi_{MOND}∼1/r$ in the Newtonian limit. This means that in the MOND regime, the force acting on test-particles orbiting a large mass M (e.g., the galactic center) is proportional to 1/r, resulting in flat rotation curves. It also follows that the total mass $M∼v^{4}$, where *v* is the limiting velocity of the rotation curves. This is the observationally well-established and important baryonic Tully–Fisher relation [11] (BFTR) (though observations allow a power in the range 3.5–4).

## 3. Lagrangian Approach

### Bekenstein Formulation(s)

The first complete theory of MOND, dubbed AQUAL, was constructed in 1984 by Milgrom and Jacob Bekenstein [10]. AQUAL generates MONDian behavior by modifying the gravitational term in the classical Lagrangian from being quadratic in the gradient of the Newtonian potential to a more general function (AQUAL is an acronym for AQUAdratic Lagrangian). AQUAL is expressed as: (8)$$\begin{array}{ccc} L_{\mathrm{Newton}} & = & -\frac{1}{8\pi G}{∥\nabla \phi ∥}^{2}\;,\\ L_{\mathrm{AQUAL}} & = & -\frac{1}{8\pi G}a_{0}^{2}F\left(\frac{{∥\nabla \phi ∥}^{2}}{a_{0}^{2}}\right)\;, \end{array}$$
where ϕ is the standard Newtonian gravitational potential and *F* is a dimensionless function such that $\mu \left(x\right)=\frac{dF(x^{2})}{dx}$. The complete AQUAL Lagrangian is: (9)$$L_{\mathrm{AQUAL}}=\rho \phi +\frac{1}{8\pi G}a_{0}^{2}F\left(\frac{{\left|\nabla \phi \right|}^{2}}{a_{0}^{2}}\right)\;.$$Applying the variational principle, Equation (9) leads to a nonlinear generalization of the Newton–Poisson equation, i.e., Equation (3). AQUAL generates MONDian behavior by modifying the gravitational term in the classical Lagrangian from being quadratic in the gradient of the Newtonian potential to a more general function.

There are departures and generalizations of MOND, e.g., Milgrom’s Quasilinear MOND (QUMOND) [12] (also derived from an action), which involves two potentials—ϕ, which dictates particle accelerations, and an auxiliary potential $\phi_{2}$. There is even a bimetric MOND gravity (BIMOND) [13] which is a class of (complicated) relativistic theories involving two metrics and two respective Ricci scalars.

Appendix B of Bekenstein’s 1984 paper considers a relativistic generalization of AQUAL, i.e., a RAQUAL Lagrangian made of three terms, one of which is a gravitational scalar–tensor theory which connects to the Brans–Dicke theory and the (now defunct) theory of Hoyle and Narlicker. The argument ${\left|\nabla \phi \right|}^{2}$ of the function *F* is replaced by a relativistic generalization $\psi_{,\nu }\psi_{,}^{\nu }$ where ψ is the scalar field of the scalar–tensor theory.

According to Sanders and McGaugh [14,15], one problem with AQUAL (or any scalar–tensor theory in which the scalar field enters as a conformal factor multiplying Einstein’s metric) is AQUAL’s failure to predict the amount of gravitational lensing actually observed in rich clusters of galaxies.

The tensor–vector–scalar (TeVeS) theory, the bellwether of relativistic MOND, was put forth by Bekenstein [16]. TeVeS solves problems associated with earlier attempts to generalize MOND, such as superluminal propagation. However, it has its criticisms. Note that the functions μ and *F* in Equation (9) are redefined according to $\mu \left(\sqrt{y}\right)=dF\left(y\right)/dy$ where ([16], Equation (4)):(10)$$F\left(y\right)\to \left\{\begin{array}{cc} y & y\gg 1\\ \frac{2}{3}y^{3/2} & y\ll 1 \end{array}\right.\;.$$Note the fractional power 3/2 which will appear in the next section. Though successful, MOND remains a curve-fitting phenomenological theory. Originally set up in a non-relativistic framework, it was consequently unsuitable for cosmology because galaxies are governed by GRT. MOND’s relativistic generalizations are plagued by theoretical difficulties [17,18]. Several ad hoc and inelegant additions to general relativity are required to create a theory compatible with a non-Newtonian non-relativistic limit, even though the predictions in this limit are rather clear.

## 4. Superfluid Lagrangian Formulation

Sabine Hossenfelder [1,19,20] uses the Lagrangian made from Khoury [21] who theorizes that DM is a Superfluid. The conjecture is that the field ϕ DM Superfluid phonons are governed by the MOND Lagrangian at T=0: (11)$$L_{\phi }=\frac{2\Lambda {\left(2m\right)}^{3/2}}{3}X\sqrt{\left|X\right|}\;,$$
where *X* is the free kinetic energy of the field ϕ, *m* is a constant of dimension mass and Λ qualifies the strength of the self-interaction. In the Newtonian limit, the kinetic term can be approximated by clarifying the gravitational potential:(12)$$X\approx \mu -m\Phi_{N}+\dot{\phi }-\frac{1}{2m}{\left(\nabla \phi \right)}^{2}\;,$$
where $\Phi_{N}$ is the Newtonian gravitational potential and μ can be interpreted as a chemical energy potential. The dot is a time derivative with respect to time. The gradient contains spatial derivatives only. It is claimed by Khoury et al. that (12) is strikingly similar to that of the Unitary Fermi Gas (UFG) which has generated much excitement in the cold atom community in recent years. Khoury et al. noted that the fractional power of *X* would be strange if (12) described a fundamental scalar field. As a theory of phonons, however, the power determines the Superfluid equation of state, and fractional powers are not uncommon. Indeed, the effective field theory for the UFG Superfluid is $L∼X^{n}$, where n=5/2 in 3+1 dimensions and 3/2 in 2+1 dimensions, and is therefore also non-analytic.

It seems the Lagrangian (11) with (12) is very much a Bekenstein-type RAQUAL with a function *F* replaced by *X* while retaining arguments such as ${\left(\nabla \phi \right)}^{2}$, the resemblance to UFG Superfluid being perhaps a bonus. Sabine Hossenfelder emphasizes [20] that:
“It is the combination of the power 3/2 in the kinetic term (11) combined with the peculiar coupling (12) that gives rise to the MOND-like behavior. The same features can be found in the vector-based model considered …” in her work on “emergent gravity” [19]. However, this same reference also returns to Bekenstein’s RAQUAL. To mediate a force between baryons, DM phonons must couple to the baryon density as: (13)$$L_{int}=\alpha \Lambda \frac{\phi }{M_{PI}}\rho_{b}\;,$$
with α being a constant, $\rho_{b}$ the baryonic density and $M_{Pl}$ the Planck mass. The term in Equation (13) is now added to Equation (11). At zero temperature, the effective theory thus has three parameters: the particle mass *m*, a parameter Λ related to the self-interaction strength and the coupling constant α between phonons and baryons. Khoury et al. continue with:
“A fourth parameter of the particles themselves is their self-interaction cross-section σ setting the conditions for their thermalization, while a fifth parameter β will later be introduced to accommodate for finite-temperature effects” [21]. One is reminded by the statement attributed to J. Von Neumann revealed in a meeting between Freeman Dyson and Enrico Fermi: “With 4 parameters, I can fit an elephant. With 5, I can make it wiggle its trunk” [22]. This brings us to the finite-T Lagrangian: (14)$$L_{\phi }=\frac{2\Lambda {\left(2m\right)}^{3/2}}{3}X\sqrt{\left|X-\beta (T)Y\right|}\;,$$
where β>3 and $Y=\mu +\dot{\phi }+v•\nabla \phi$, as required for stability.

The reported justification for the Lagrangian [21] is as follows: $L_{\phi }+L_{int}$ from Equations (11)–(13) give rise to the MOND force law. Assuming a static profile, the phonon equation of motion is then:(15)$$\nabla •\left(\frac{{\left(\nabla \phi \right)}^{2}-2m\hat{\mu }}{\sqrt{{\left(\nabla \phi \right)}^{2}-2m\hat{\mu }}}\nabla \phi \right)=\frac{\alpha \rho_{b}}{2M_{Pl}}\;,$$
where $\hat{\mu }=\mu -m\Phi$ and which looks like our Poisson equation of Equation (3). Khoury et al. then proceed to obtain the critical acceleration $a_{0}$ to realize, amongst other things, that they need to use the temperature-dependent result of (14). Their work [21] seems to address a number of points concerning MOND and the DM Superfluid model. Khoury’s work cites the analysis of Bruneton et al. [23], whose abstract states:

“Our conclusion is that all MOND-like models proposed in the literature, including the new ones examined in this paper, present serious difficulties: Not only they are unnaturally fine tuned, but they also fail to reproduce some experimental facts or are unstable or inconsistent as field theories. However, some frameworks, notably the tensor-vector-scalar (TeVeS) one of Bekenstein and Sanders, seem more promising than others, and our discussion underlines in which directions one should try to improve them.”

In a nutshell, it seems the gravitational scalar–tensor theories examined were problematic, and although TEVES seemed most promising, it is not perfect either. As for a further elaboration about the problems with MOND, see, e.g., Ref. [24].

### Relativistic Completion

Interestingly, Khoury et al. [25] attempt to derive the Lagrangian of (11) and (12) using more fundamental Lagrangians. Their appendix points out that: $$L=-|\partial_{\nu }{\Phi |}^{2}-m^{2}{\left|\Phi \right|}^{2}-\frac{\lambda }{3}{\left|\Phi \right|}^{6}\;,$$
gives $L\left(X\right)\approx X^{3/2}$ of (11) but with the wrong sign. Instead, they consider: (16)$$L=-\frac{1}{2}\left(|\partial_{\nu }{\Phi |}^{2}+m^{2}{\left|\Phi \right|}^{2}\right)-\frac{\Lambda^{4}}{6(\Lambda_{c}^{2}+{\left|\Phi \right|}^{2}{)}^{6}}{\left(|\partial_{\nu }{\Phi |}^{2}+m^{2}{\left|\Phi \right|}^{2}\right)}^{6}\;,$$
where “the scale $\Lambda_{c}$ is introduced to ensure that the theory admits a Φ=0 vacuum”. Khoury et al. did their best to reconcile MOND and one of the better relativistic DM models, namely ΛCDM. Unfortunately, the Lagrangian (16) is very unappealing: the formulation looks complicated and contrived. (Moreover, there is apparently a missing factor of 2 in going from their Equation (21) back to a non-relativistic counterpart in their Equation (22) of Ref. [26]). Equation (16) looks complicated even as it has only basic kinetic and mass terms. Khoury ([25], [1.1]) reports that MOND fails at large extra-galactic scales. Some of the criticism of this approach is best expressed by Sabine Hossenfelder [27]:

“The biggest problem is that it’s not very well understood under exactly which conditions DM forms a Superfluid. There are also different kinds of particles that can form a Superfluid and it’s not clear which of those fits the data best. Another problem is that it’s really not well understood how a fluid condenses to a Superfluid in a curved space-time. That’s because the people who normally study Superfluids don’t have to think about gravity all that much. If they take it into account at all, it’s a vertical gradient in the laboratory…I think it’s a mistake to regard DM and MOND as two competing theories, each of which has to be made to fit all of the data. To me the data say the answer is a combination of both.”

The trouble is that MOND and relativistic CDM models cannot be reconciled [28]. We need an alternative.

## 5. Alternative—LogSE Formulations

Clearly all this material concerning MOND and DM presented here, and attempting to reconcile or combine them, involves “mature industries”. For example, the very appearance of artificial interpolating functions μ(x) or F(x) in present MOND theories indicates that none are the basic MOND theory we are after, i.e., a “FUNDAMOND” theory. The existing MOND theories are, at best, only effective, approximate theories of limited validity. MOND is an “effective” theory, useful indeed, but artificial, and the attempts at putting it on a firm foundation have been less than perfect. Mind you, we retain the notion that a gravitation Lagrangian might do the trick. This is feasible as long as we keep it simple and “natural”. We start from a free Lagrangian formulation for *dilatonic gravity* [29,30]: (17)$$\int d^{4}xL_{F}=\int d^{4}x\sqrt{-g}\left\{\Psi R+\frac{1}{2}g^{\mu \nu }\nabla_{\mu }\Psi \nabla_{\nu }\Psi +2\Lambda \right\}\;,$$
where 8πG=1 and Ψ is the dilaton field which ensures the correct Newtonian limit of (17) in *d* (spatial) dimensions where d=1,2,3. Equation (17) is a special case of a general class of relativistically invariant scalar–tensor formulations, an F(R) theory that has been loop-quantized [29,31] (although we stop short of using the Ashtekar variables). Equation (17) embeds a property of conformal invariance, and when Ψ=1, it reduces to the limit of GRT. At any rate, a relativistic version of the LogSE can be obtained by replacing its Laplacian with the d’Alembertian, similarly to the Klein–Gordon equation [32].

We applied the variational principle using the Arnowitt–Deser–Misner (ADM) method [33] to Equation (17) with a particular choice of gauge and coordinate conditions while assuming no transverse-traceless metric components, the metric $g_{ab}$ reduces to the 3×3 *isotropic* form, i.e.,
(18)$$g_{ab}=\gamma_{ab}=\delta_{ab}𝔥\left(1+\frac{1}{2}h^{T}\right)\;\to \;\sqrt{\gamma }={𝔥}^{3/2}\;,$$
where $h^{T}$ corresponds to the trace of the transverse part of the metric and carries the asymptotic (Newtonian) 1/r part of the metric such that [33]
$$\lim_{r\to \infty }h^{T}=0\;.$$With the following transformation,
(19)Ψ(t,x,y,z)=F(t)−5ln(|ψ(x,y,z)|).
Defining Φ=1/4ψ and F(t)=F is a constant in time, it is found that the field equation governing the transformed dilaton field Φ is given by: (20)$$-\frac{1}{2}\nabla^{2}\Phi +V\Phi +S\Phi \ln\left(|\Phi |\right)=E\Phi \;,$$
where E=E(Λ) and $S=S({𝔥}^{3}R)$ and ${}^{3}R$ is the three-dimensional Ricci scalar. Equation (20) is an *energy-balancing* equation for the dilaton field which is governed by a *logarithmic Schrödinger equation* (LogSE) with *E* as a function of the cosmological constant acting as the eigenvalue. The energy balance in itself is more attractive than the previous formulations leading to Equation (16), which involve mass and kinetic energy only. In flat space where the Ricci scalar is zero, Equation (20) reduces to the standard Schrödinger equation.

External potentials can be added to *V* in Equation (20). The LogSE already has a number of fundamental applications ranging from quantum gravity to nuclear physics to magma transport to information theory (e.g., see [34] and the references therein). For example, it can provide the upside-down “Mexican hat” shape of the effective potential for the Higgs boson which is different from the one used in the Glashow–Weinberg–Salam model, yet it yields the mass generation and it is free of the imaginary-mass problem appearing in the conventional Higgs potential [35]. (In principle, the Higgs boson is distinct from the gravitational dilaton because they result from different theories, but the apparent similarity between these two particles has made some wonder if they are related to each other (e.g., see [36]), but this is not pursued here). Other examples of the LogSE concern Bose–Einstein condensates and Superfluids.

Because Superfluid phonons were previously mentioned, it is worth noting that precise experimental data show that at temperatures below 1 K, the pressure in (liquid) Helium-4 has a cubic dependence on density and, thus, the speed of sound $c_{s}$ scales as a cubic root of pressure:(21)$$c_{s}=K^{1/3}{\left(P-P_{c}\right)}^{\nu }\;\mathrm{where}\;\nu =\frac{1}{3}\pm 0.01,$$
where $P_{c}$ is the critical pressure.

Near this critical pressure point, this speed approaches zero, whereby the critical pressure is negative, thus indicating a cavitation instability regime. Figure 1 is a reproduction from Ref. [37] and resoundingly confirms Equation (21). To explain this dependence, one had to model the liquid helium as a mixture of three quantum Bose liquids:Dilute (Gross–Pitaevskii-type) Bose–Einstein condensate;Ginzburg–Sobyanin-type fluid;Logarithmic Superfluid.This was encapsulated by the quantum wave equation:(22)$$\left[-i\hbar \;\partial_{t}-\frac{\hbar^{2}}{2m}{\vec{\nabla }}^{2}+V_{ext}\left(x,\;t\right)+{F(|\Phi |}^{2})\right]\Phi =0,$$
which derives from the Euler–Lagrange equation from the action functional on the Lagrangian.
$$L=\frac{i\hbar }{2}\left(\Psi \partial_{t}\Phi^{*}-\Phi^{*}\partial_{t}\Phi \right)+\frac{\hbar^{2}}{2m}{\left|\nabla \Phi \right|}^{2}+V_{ext}\left(r,t\right)+{V(|\Phi |}^{2})\;,$$
where F=∂V(ρ)/∂ρ is found to be ([37], Equation (14)):(23)$$F\left(\rho \right)=-\epsilon \left[\alpha^{2}\ln\left(\rho /\bar{\rho }\right)+2\frac{\alpha }{3}\frac{\rho }{\bar{\rho }}+\frac{1}{18}\frac{\rho^{2}}{{\bar{\rho }}^{2}}\right]\;,$$
where α is a constant. The logarithmic term in F(ρ) of Equation (23) is essential to realize Figure 1. Without it, one would obtain at best a proportionality $c_{s}^{4}\propto \left(P-Pc\right)$ rather than the experimentally vindicated $c_{s}^{3}\propto \left(P-Pc\right)$. Bear in mind that it is the logarithmic part of (23) that dominates the asymptotic behavior of the wave function Φ in various regimes. A similar vindication can also be found for cold sodium atoms [38]. Thus, the LogSE deriving from a general relativistic formulation is versatile and precise in applications to Bose–Einstein condensates and Superfluids and is consequently a good candidate for modeling galactic rotation curves. On this issue, the LogSE in the context of the log BEC Superfluid Vacuum Theory (SVT) provides us with an alternative to DM [39,40]. Assuming spherical symmetry and updating *F* of the LogSE in Equation (22) with:(24)$${F(|\Phi |}^{2})\to \left(b_{0}-\frac{q}{r^{2}}\right)\ln\left(\frac{{\left|\Phi \right|}^{2}}{\bar{\rho }}\right)\;.$$
We inject an ansatz for Φ for the vacuum in the form ([39], Equation (5)):(25)$$|\Phi |\;=\;\sqrt{\bar{\rho }}{\left(\frac{r}{\ell }\right)}^{\chi /2}P\left(r\right)\exp\left(-\frac{a_{2}}{2}r^{2}+\frac{a_{1}}{2}r+\frac{a_{0}}{2}\right)\;,$$
where P(r) is a polynomial in *r* and the exponential term is a *Gausson,* i.e., a single soliton which often appears as a ground-state solution to the LogSE [34,41]. It is found that for central potential problems, the solution of the LogSE can be well approximated by linear combinations of such functions as in (25) [34,42], justifying the ansatz. The injection of this ansatz into the logarithmic part of the LogSE, i.e., $F(|\Phi {|}^{2})/m$, generates six individual potential terms ([39], Equations (6)–(12)) of which two have the form:(26)$$\begin{array}{ccc} V_{N} & = & -\frac{a_{1}q}{m}\frac{1}{r}=-\frac{GM}{r}\;,\\ V_{ln} & = & \frac{b_{0}}{m}\chi \left(\ln\left(\frac{r}{\ell }\right)+\ln P^{2}\left(r\right)\right)\;. \end{array}$$Thus, we recover both the standard Newtonian potential and the MOND logarithmic potential of Equation (1). The log term of the LogSE naturally provides the MOND logarithmic potential. Equation (26), with the remaining four potential terms and the coefficients of these terms, can be successfully fitted to a number of astrophysical cases.

This alternative can address cases where the conventional models fail. For example, Corbelli et al. show results where the outer regions of galactic rotational curves start to significantly deviate from predictions of popular theories like MOND [43,44]. Granted, the alternative does produce a number of parameters for fitting as in the earlier models. However, the more conventional Superfluid dark matter models also use ordinary matter distributions, giving them a large freedom for fitting, and yet they have been confronted with underfitting or overfitting. In the alternative here, the parameters start from a simpler formalism and a simple ansatz. The parameters are products of the fixed parameters of the LogSE model but also parameters of the “local” vacuum’s wave function which can vary from galaxy to galaxy, thus providing a sufficiently large domain of parameters that is flexible yet with a reduced danger of underfitting or overfitting.

On a galactic scale, the model of Ref. [39] explains the non-Keplerian behavior of galactic rotation curves and also why their profiles can vary depending on the galaxy. This approach is further vindicated in modeling the asymptotic behavior of these galactic rotation curves [40] and can even model galaxies for which the rotation velocity profiles do not have flat regions [45]. Thus, the LogSE models of a log BEC SVT presented here:Have a connection to a relativistic gravity theory;Have a proven capacity for modeling BECs and cold atoms;Involve a much simpler Lagrangian formulation than those of the previous Superfluid models;Recover the advantages of MOND for modeling galactic rotation curves.Concerning the first item, caution should be exercised here. The LogSE can be obtained by various routes in different physical scenarios. It does not follow that log BCE SVT necessarily involves dilatonic gravity or vice versa because both have a LogSE in common. However, note that the log SE in Equation (20) obtained from dilatonic gravity is perhaps the first LogSE formulation with a *non-constant* coefficient for the logarithmic term and that the $1/r^{2}$ term (with *q* as coefficient) of Equation (24) is essential to recover the Newtonian limit inside a galaxy which is the Ricci scalar for a 3D sphere. The latter equation and its ansatz in Equation (25) both use spherical symmetry. (This is also rather tantalizing but will not be pursued here).

Moreover, without actually denying the existence of DM and/or DE, the very hypothesis of their existence is not needed here. The overhead and the complications resulting from such a hypothesis are therefore bypassed. Our Lagrangian formulations are for Bose–Einstein condensates, and it is doubtful they could model DM at a microscopic level because fermions govern the atomic and molecular structure. (Also note that baryons are also fermions!) Furthermore, the modeling of an entire galaxy with a fluid or a Superfluid already involves outrageous assumptions just from the point of view of continuity. Dealing with the the baryon density, e.g., in the RAQUAL theory, is a burdensome overhead from either a conceptual or pragmatic point of view.

### Addressing the Objections to Nonlinear Quantum Formulation

Some, like the authors of Ref. [46], claim the only consistent way to manipulate quantum amplitudes consistently is with a standard *linear* Schrödinger equation. Thus, they would conclude that the LogSE is not a valid quantum-mechanical formulation.

However, the log nonlinearity in the LogSE merely describes many-body effects because the log term $\ln(|\Phi {|}^{2})$ is the Everett–Hirschman entropy [47]. The log nonlinearity occurs when the collective phenomenon, such as a Bose–Einstein condensate, emerges with its own degrees of freedom. In other words, one should not confuse its solution Φ with the wave function of the linear yet fundamental standard quantum mechanics. A general justification for using log nonlinearities for many-body and open quantum systems is given in Ref. [47].

## 6. Conclusions

From Modified Newtonian Dynamics (MOND) to Superfluid dark matter (DM) models to the alternative based on the log BEC Superfluid Vacuum Theory (SVT), we have progressed from a curve-fitting phenomenological theory that is MOND to a fundamental formulation for BECs, cold atoms and even galactic rotation curves, based on the logarithmic Schödinger equation (LogSE). However, the nonlinear aspects of the LogSE do present a challenge.

Although not strictly quantum mechanical, the LogSE has many properties in common with the linear Schrödinger equation. In spite of the nonlinearity, wave mechanics can be established [48], complete with discrete solutions and orthogonality conditions between them. As mentioned before, linear combinations of Gaussons can well approximate the solutions of the LogSE. This is explained by a *quasi nonlinear property of superposition* which has been firmly established for harmonic potentials [49]. The precise numerical work on the LogSE for hydrogenic systems [34,42] suggests that the LogSE is even amenable to computational quantum chemistry while also allowing for soliton solutions, which is intriguing. As suggested by Equation (25), Gaussons can be viewed as a product of a Slater-type (hydrogenic) function (e.g., of the form $\exp(-a_{1}*r/2)$) and a Gaussian (e.g., of the form $\exp(-a_{2}*r^{2}/2)$). Note that Slater functions and especially bases of Gaussian functions have provided a proven reliable mathematical computational technology for atomic and molecular physics as well as quantum chemistry [50,51,52]. There are some differences though: solutions to the LogSE can be square integrable, but the logarithmic term retains the normalization constant.

From a physical point of view, we cannot reject nonlinear Schrödinger equations like the LogSE. They are needed for Superfluids and Superconductors. Who could deny that the zero-resistivity property of Superconductors or the zero-viscosity property of Superfluid at supremely cold temperatures to be anything but bonafide *quantum effects?* These have no classical explanation. In particular, standard quantum mechanics applies especially well at cold temperatures where thermodynamic effects are not overwhelming.

It has been said that it is perhaps too much to expect that a fundamental equation for describing nature be linear after all. Moreover, it is not surprising that the transition from a (nonlinear) GRT action with a dilaton field in Equation (17) would lead to a nonlinear equation governing the dilaton field, i.e., Equation (20). Equation (19) infers Φ∝exp(constant∗Ψ) where Ψ is the dilaton field, suggesting an intriguing interpretation for the wave function Φ, one based on *differential geometry*. In trying to reconcile the GRT with standard quantum mechanics, a nonlinear wave equation corresponding perhaps to a different or intermediate regime might be the gateway. 

## Figures and Tables

**Figure 1 entropy-25-00012-f001:**
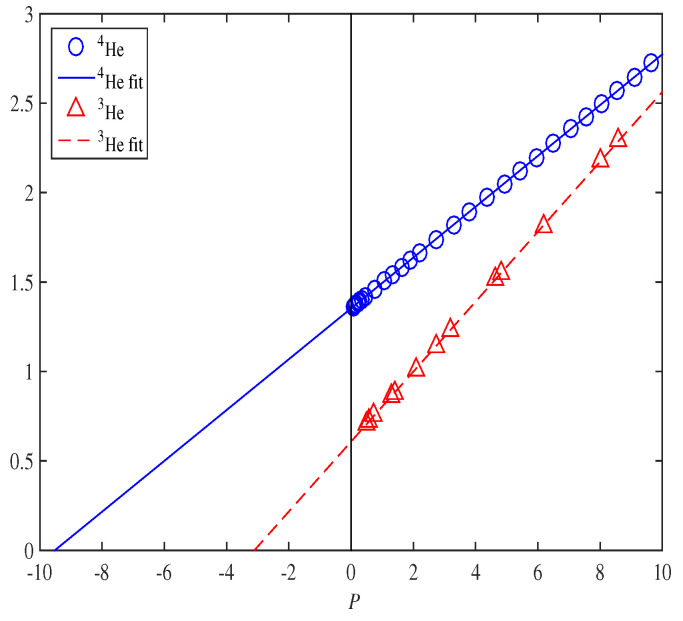
Profiles of $c_{s}^{3}$ in units of $10^{13}{\mathrm{cm}}^{3}/s^{3}$, versus pressure *P*, in atm from precise experimental data and fitted for ${}^{4}$He (circles, solid fitting curve) and ${}^{3}$He (triangles, dashed fitting curve) in the regime 0<P<10 bars. (Courtesy Ref. [37]).

## Data Availability

Not applicable.

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
