# Peer review of "From Modified Newtonian Dynamics to Superfluid Vacuum Theory"

_entropy, 2022, doi:10.3390/e25010012_

Round 1

Reviewer 1 Report

The article is devoted to the review of Modified Newtonian Dynamics and the hypothesis that dark matter is perhaps a BEC described by the Logarithmic Schödinger equation.

Although the review of MOND is not complete, it is nevertheless sufficient for comparison with the BEC-based dark matter model described by the Logarithmic Schödinger equation. This model is a very interesting candidate for explaining the nature of dark matter, since the Logarithmic Schödinger equation has interesting and unusual physical and mathematical properties.

There are some typos in the text of the article that should be corrected:

1. In formula (1), there is an extra closing bracket.

2. In line 89, the number of the formula is given without brackets.

3. In lines 98, 124 and before formula (13), there is a typo: Berkenstein

4. In lines 184 and 266, a dot is absent.

In this connection, I believe that, upon correcting the typos, this article may be published in the Special Issue "Completeness of Quantum Theory: Still an Open Question".

Author Response

see attached pdf file

Reviewer 2 Report

The paper was written more as a review than the original paper. However, it still delivers very interesting results than can warrant the publication of the paper in the journal of Entropy. In spite of that, I have some minor comments that can easily be addressed. These are the followings:

1) In eq. (1), there is extra ) on the right side, that should be removed.

2) Introduction of the paper is too short for the topic presented in the paper. It should be extended.

3) Most of the formula or expressions presented in the review parts should be shown with brief derivations.

4) To show behaviors of the obtained physical quantities, at least some figures should be presented.

Author Response

see attached pdf file

Reviewer 3 Report

In the second paragraph of "introduction", the claim that the MOND theory has been very successful in confrontation with observations seems a little bit stronger. To my knowledge, there are still some controversies remained. I suggest the author may give more discussions on the observational aspects, or at least, list more relevant references.

Author Response

see attached pdf file

Reviewer 4 Report

Report on the paper "From Modified Newtonian Dynamics to Superfluid Vacuum Theory" by Tony C. Scott

==================================

The paper discusses some issues of the MOND and superfluid DM approaches to explain galactic rotation curves. After some critical analysis, the author considers an alternative scheme based on the superfluid vacuum theory with a nonlinear logarithmic Schrodinger equation (LogSE), which may reconcile two approaches mentioned above. This manuscript can be considered as a review paper briefly describing current issues in the MOND-motivated theoretical framework, as well as in the superfluid-based schemes. The author emphasizes more fundamental character of LogSE approach as compared to the MOND and Superfluid DM models supposed to be mostly "effective" and phenomenological. At the same time, in my view, the LogSE approach is also not free of phenomenological aspects being based on the specific ansatz with a number of fitting parameters. Eventually, the modified logarithmic Schrodinger equation can hardly be regarded as a fundamental one. It is a result of an effective phenomenological description of systems with complicated structure of interactions, but on the fundamental level their description must be based on the standard quantum mechanical postulates where the linear superposition principle is a core of the entire formalism. Possible non-linearity can arise in nature on the fundamental level from the quantum field theory, including non-abelian Yang-Mills theories, curved background, loop corrections and quantum gravity effects, but none of these points seem to be directly connected with the LogSE approach. But this is my view rather than a significant critical remark.

I think the paper is an interesting review and can be published. I only recommend to carefully check the text of the manuscript, because some places seem to be confusing, for example:

Strange bracket formatting in Eq. (1);

$f (y)$ of $F(y)$ in Eq.(10)?

There is no $\Phi$ in Eq. (11), so which $\Phi$ is mentioned in the phrase "where $\Phi$ is the gravitational potential" after Eq. (11)?

The same for the phrase "where $\Phi$=" after Eq. (19);

What is $h^T$ in Eq. (18)?

$V(|\Phi|^2)$ or $F(|\Phi|^2)$ in the non-numbered equation between Eqs. (22) and (23)?

The grammar style and typos should also be checked.

-------

To conclude, I recommend the paper for publication in its present form (after careful proofreading and eliminating misprints).

Author Response

see attached pdf file
